# Improving Stability of Roll-to-Roll (R2R) Gravure-Printed Carbon Nanotube-Based Thin Film Transistors via R2R Plasma-Enhanced Chemical Vapor-Deposited Silicon Nitride

**DOI:** 10.3390/nano13030559

**Published:** 2023-01-30

**Authors:** Sagar Shrestha, Sajjan Parajuli, Jinhwa Park, Hao Yang, Tae-Yeon Cho, Ji-Ho Eom, Seong-Keun Cho, Jongsun Lim, Gyoujin Cho, Younsu Jung

**Affiliations:** 1Department of Biophysics, Institute of Quantum Biophysics, Research Engineering Center for R2R Printed Flexible Computer and Department of Intelligent Precision Healthcare Convergence, Sungkyunkwan University, Suwon-si 16419, Republic of Korea; 2Thin Film Materials Research Center & Chemical Materials Solution Center, Korea Research Institute of Chemical Technology (KRICT), Daejon 34114, Republic of Korea

**Keywords:** roll to roll, PECVD, passivation, threshold voltage, silicon nitride, 85/85 test

## Abstract

Single-walled carbon nanotubes (SWCNTs) have an advantage in printing thin film transistors (TFTs) due to their high carrier mobility, excellent chemical stability, mechanical flexibility, and compatibility with solution-based processing. Thus, the printed SWCNT-based TFTs (pSWCNT-TFTs) showed significant technological potential such as integrated circuits, conformable sensors, and display backplanes. However, the long-term environmental stability of the pSWCNT-TFTs hinders their commercialization. Thus, to extend the stability of the pSWCNT-TFTs, such devices should be passivated with low water and oxygen permeability. Herein, we introduced the silicon nitride (SiNx) passivation method on the pSWCNT-TFTs via a combination of roll-to-roll (R2R) gravure and the roll-to-roll plasma-enhanced vapor deposition (R2R-PECVD) process at low temperature (45 °C). We found that SiNx-passivated pSWCNT-TFTs showed ± 0.50 V of threshold voltage change at room temperature for 3 days and ±1.2 V of threshold voltage change for 3 h through a Temperature Humidity Test (85/85 test: Humidity 85%/Temperature 85 °C) for both p-type and n-type pSWCNT-TFTs. In addition, we found that the SiNx-passivated p-type and n-type pSWCNT-TFT-based CMOS-like ring oscillator, or 1-bit code generator, operated well after the 85/85 test for 24 h.

## 1. Introduction

Single-wall carbon nanotubes (SWCNTs) are highly promising as semiconducting materials for printing thin film transistors (TFTs) due to their unique mechanical, electrical, and chemical properties [1,2]. Furthermore, thanks to their specific chirality (semiconducting or metallic characteristics) [3], the mixed semiconducting and metallic SWCNTs are fascinating candidates as active materials in printed TFTs because of their low cost and robustness. Therefore, the printed SWCNT-based TFTs (pSWCNT-TFTs) are highly appropriate to integrate logic circuits or TFT active matrix-based sensors [4,5,6,7]. Various printing methods have been employed to develop flexible, light, and costless pSWCNT-TFT-based devices [8]. In general, the back-gated pSWCNT-TFTs exposed the SWCNT channels to surroundings (water and oxygen), resulting in unstable electrical performance. For example, when the electrical properties of pSWCNT-TFTs were measured in a vacuum, the on currents were lower than those measured in ambient conditions [1]. Moreover, since the back-gate pSWCNT-TFTs have typically unipolar p-type properties, it seems difficult to create n-type SWCNT-TFTs. However, the n-type SWCNT-TFTs are indispensable in pSWCNT-TFTs because it allows pSWCNT-TFT-based complementary metal–oxide–semiconductor (CMOS)-like logic gates and analog circuits to operate with lower power [9,10,11]. In recent years, we have reported that the conversion from p-type to n-type SWCNTs can be achieved by simply printing n-doping ink on printed p-type SWCNTs via a roll-to-roll (R2R) gravure or roll-to-plate (R2P) printing methods [9,12]. However, unfortunately, as soon as one exposes the n-type SWCNT-TFT to the surrounding environment, it starts to slowly revert to a p-type one, causing device failures.

In pSWCNT-TFTs, especially for n-type pSWCNT-TFTs, instability is caused by two main reasons: electronic ink impurities and environmental factors. Electronic ink impurities can be removed by selecting better materials, including polymer binders. However, after removing the impurities, environmental factors such as water, oxygen, and light are inevitable during the operation of the device in ambient conditions by generating the trapped charges at the interface between the dielectric and the SWCNT semiconducting layers. Moreover, such trapped charges are generated via electron transfer from the SWCNT to water/oxygen due to the redox potential difference between water/oxygen and the fermi level of the SWCNT [13,14]. Since the valence band of the SWCNTs lies close to the redox potential of the oxygen-dissolved water, the electrons can transfer easily from the SWCNT to the oxygen/water layer, causing large hysteresis and instability in the pSWCNTs. Therefore, to prevent such charge transfer from the SWCNT and make devices robust against environmental factors, a dense passivation method is needed to isolate the printed device from water and oxygen molecules.

Regarding the passivation of pSWCNT-TFTs, many research groups have reported methods to provide device stability. Zhang et al. reported polymer-based passivation using SU-8 and polymethyl-methacrylate (PMMA) with thicknesses of 2 µm and 200 nm for pSWCNT-TFTs. However, when the SU-8- and PMMA-passivated samples were tested under nitrogen conditions, the diffusion through the layer within 1 h caused variations in devices [15]. Similarly, Dai et al. have also passivated SWCNT-TFTs using PMMA [16], but water molecules still affect the TFTs’ performance through the PMMA layers. Furthermore, Kamim et al. and Kim et al., respectively, employed Si_3_N_4_ via chemical vapor deposition (CVD) at 270 °C and Al_2_O_3_ deposited via atomic layer deposition (ALD) at 300 °C for showing stable SWCNT-based devices [3,16]. Similar observations were made by Qu et al., who passivated sputtered SnO channels by Al_2_O_3_ thin films using the ALD method to protect the degradation of p-type SnO-based TFTs [17]. Even though they attained stable performance, high temperatures cannot be used to passivate flexible substrates such as polyethylene terephthalate (PET) or polyethylene naphthalate (PEN) film. In addition, previously, our group reported stable performance by printing the passivation layers of CYTOP and FG-3650 on the top of the pSWCNT-TFT-based devices, which showed stable output signals using six layers of passivation [9]. However, the devices became unstable when continuous bias stress was applied [9]. F. Iqbal et al. reported alleviating the degradation under bias stress by depositing perylene over the device [18]. Even though the method showed stable characteristics of the device under ambient conditions, it is hard to apply in the R2R-printed devices directly. Since the R2R-printed devices should be passivated with a continuous in-line deposition method, we need to integrate an R2R passivation method with the R2R printing system. Since the R2R-printed flexible electronic devices should experience minimized exposure time to surroundings, the passivation method should be connected with the R2R printing system. Among available methods to integrate with the R2R printing system, a plasma-enhanced chemical vapor deposition (PECVD) method has been widely used to provide high-quality passivation layers. However, since PECVD is fabricated by the collision between energetic electrons and molecules of source gases, induced defects or extra-doping on pSWCNTs would often be observed. 

Here, we report R2R PECVD of SiNx on the R2R-printed p-type and n-type SWCNT-TFTs via primary coating of CYTOP to prevent defects and extra-doping. Since silicon nitride (SiNx) shows less permeability towards the water and oxygen material [19], deposited SiNx layers will effectively prevent the further degradation caused by water and oxygen molecules. Observing the variation of threshold voltage (∆Vth) as a function of time, the SiNx-passivated pSWCNT-TFTs showed a very small ∆Vth. When the inverter test was also carried out by giving continuous electrical bias stress to the pSWCNT-TFTs for 1 hr, a minimal shifting of output signal was observed. The passivated pSWCNT-TFTs were also monitored through a Temperature Humidity Test (RH: 85%/85° C), where the ∆Vth of ±1.2 V was observed for 3 h. Furthermore, the R2R-passivated pSWCNT-TFT-based CMOS-like ring oscillator, called a 1-bit code generator, was stable even under 85/85 test conditions for 24 h. Those results showed a way of providing a long lifetime of R2R-printed SWCNT-TFT-based devices for practical applications.

## 2. Materials and Methods

### 2.1. Ink Formulation

For the fully R2R gravure printing process, Ag nanoparticle-based ink was formulated by following our previous reports [10,20]. The viscosity of Ag nanoparticle ink was adjusted by adding the diethylene glycol solvent (99%, Sigma Aldrich, St. Louis, MO, USA) to meet a viscosity of 1200 cP for the gate layer and a viscosity of 1900 cP for drain/source layer printing. For printing of the dielectric layer, BaTiO_3_ dielectric ink with 80 cP of viscosity and 32 mM/m of surface tension was further formulated based on a previously reported study [10]. For printing p-type and n-type semiconducting layers, SWCNT inks were formulated based on a reported method to meet a viscosity of 10 cP [10]. The branched poly(ethylene imine) (Sigma-Aldrich, Mw 25,000) was formulated for the n-doping layer to make the n-type SWCNT-TFTs.

### 2.2. R2R Gravure Printing and R2R PECVD Process

Ag layers for the gate layer were printed on PET web with a width of 250 mm and a thickness of 100 µm (AH71D, SKC, Korea) through the R2R gravure printing system (i-PEN, Korea) with two printing units. After printing each step, each layer was dried under 150 °C for 5 s. After completing all R2R printing processes, a roll of printed p-type and n-type pSWCNT-TFTs was moved into the R2R PECVD machine to complete the passivation step. Here, due to the collision issue between energetic electrons and molecules of source gases, the defects and extra-doping on the SWCNT layer could be induced by the plasma. Thus, the single CYTOP layer (spin coating, 500 rpm for 60 s) was introduced in this study before carrying out the R2R PECVD process. For continuously carrying out the R2R process, a pilot-scale R2R PECVD system (Roth & Rau AG, Roll Coat 500 system) was used on the R2R-printed pSWCNT-TFT roll. For fabricating the SiNx passivation layer, SiH_4_ and NH_3_ were used as the reaction gasses, and the total gas flow rate was maintained at 930 standard cubic centimeters per minute (sccm). The gas flow ratio of NH_3_/SiH_4_ was maintained at 2.65:1, while N_2_O/SiH_4_ gas flow ratio was maintained at 1.5:1. Argon (Ar) was used as a diluent gas in both cases. The base and process pressure were less than 9 × 10^−4^ and 8 Pa, respectively. To deposit a 200 nm thick SiNx passivation layer, 1500 W-powered microwave plasma and 88 mm/s web speed was used. Regarding the R2R PECVD process, more details of the optimization process for the SiNx passivation layer can be found in our previous works [21,22,23].

### 2.3. Environmental Test Conditions and Characterization

To analyze the pSWCNT-TFTs’ stability under a Temperature Humidity Test, the pSWCNT-TFT samples were stored in a chamber with 85 °C temperature/85% relative humidity (RH) conditions. These test conditions were adopted because they are usually adopted in the semiconductor and display fields for testing environmental stability (International Electrotechnical Commission (IEC) 60,068). The surface tension and viscosity of the formulated inks were measured at a constant temperature of 25 ± 2 °C using a SV-10 Vibro viscometer (A&D Co., Tokyo, Japan). In addition, a semiconductor parameter analyzer (Keithley 4200, Solon, OH, USA), a function generator (AFG 3022, Tektronix, Beaverton, OR, USA), a digital phosphor oscilloscope (DPO 4034, Tektronix, Beaverton, OR, USA), and an inductance capacitance resistance (LCR) meter (4284A, Hewlett Packard, Palo Alto, CA, USA) were used to characterize and test the pSWCNT-TFTs under normal ambient condition and a Temperature Humidity Test condition. In addition, the surface morphology of the pSWCNT-TFTs was studied using a surface profiler (NV-220, Nanosystem, Daejeon, Korea) and a microscope (semiconductor inspection microscope MX51, Olympus, Tokyo, Japan).

## 3. Results and Discussion 

For providing long-term stability of p-type and n-type pSWCNT-TFTs, water/oxygen molecules should be blocked from contacting the SWCNT layers. Otherwise, the ∆Vth of pSWCNT-TFTs will be large, primarily due to the donation of electrons to the semiconducting layer by adsorbing water and oxygen molecules (Figure 1a). Therefore, to minimize the ∆Vth, an efficient passivation layer should be applied on top of the R2R pSWCNT-TFTs by using an R2R PECVD process with the R2R gravure printing system (Figure 1b).

The p-type and n-type pSWCNT-TFTs were fabricated on PET film using an R2R gravure printing system (Appendix A). Starting from gate layer printing, continuously, the BaTiO3-based dielectric layer, SWCNT-based semiconducting layer, drain/source layers, and n-doping layer were printed with a web tension of 5 ± 0.3 kgf. First, gate layers were printed on a PET substrate using Ag nanoparticle-based ink. The detailed stage of the R2R gravure printing process and ink conditions can be found in our previous works [9]. The average thickness of the printed gate layer was 450–550 nm with 50 nm of surface roughness. After printing the gate layer, BaTiO_3_-based dielectric ink was printed on the printed Ag gate layer with a thickness of 2.0 µm. Then, SWCNT-based semiconducting ink and Ag nanoparticle-based ink for drain/source layers (~1.3 µm) were continuously printed on the dielectric layer to complete the pSWCNT-TFTs, as shown in Appendix A. As a final R2R printing step, n-doping ink was printed on SWCNT semiconducting layers to convert p-type pSWCNT-TFTs into n-type pSWCNT-TFTs. The R2R-printed n-doping layer had a thickness of 600 nm, as shown in Appendix A. 

Following the fully R2R gravure printing process, the printed p-type and n-type SWCNT-TFTs were passivated using the R2R PECVD system (Appendix A). It has a process chamber and a load lock chamber. The process chamber has the main drum where samples are rolled during the deposition process, and the plasma source is used to evaporate the material on samples. In our work, R2R PECVD was used to deposit the SiNx on the p-type and n-type pSWCNT-TFTs with 200 nm thickness (See Appendix A). When the thickness of SiNx is over 300 nm, it does not improve further, reaching some critical points, as pointed out in our previous report [23]. Corresponding to this result, SiNx with a thickness of 200 nm gives 0.0053 g/m^2^ of day vapor transmission rate (WVTR). An outer bending fatigue test was performed to examine the possibility of cracks on SiNx films as a mechanical stability test. SiNx was coated with different thicknesses on a PET substrate. Then, an outer bending fatigue test was conducted with a radius of 10 mm, under a frequency 1 Hz, and 10,000 cycles. Among different thicknesses of SiNx layer, 300 and 400 nm showed bending failure after 3000 and 1000 cycles, respectively, whereas 200 and below 200 nm completed the test without failure, as shown in Appendix A. That is why 200 nm was selected for the passivation. Thus, if the thickness is greater than this, it can cause cracks on the surface [23]. Our results and previous studies on the passivation of pSWCNT-TFTs showed a good interface quality between the SWCNT random network and the passivation layer, which can significantly impact the device’s performance [9]. The deposition temperature of the R2R PECVD process was maintained below 45 °C to be compatible with the flexible substrate. 

From the optical image of R2R gravure-printed and R2R PECVD-passivated p-type and n-type pSWCNT-TFTs, we found out structural dimensions: W/L = 1920 µm/160 µm (p-type) and W/L = 1900 µm/180 µm (n-type), as shown in Figure 2a. The thickness of the passivation layer can be controlled by adjusting the web speed and the number of repetitions of deposition. Two continuous magnetron microwave generators with a frequency of 2.46 GHz were used to generate the plasma. The deposition temperature of the SiNx passivation layer was maintained at lower than 45 °C to avoid direct damage to the printed samples. The transfer characteristics of p-type and n-type pSWCNT-TFTs with/without SiNx passivation layer (200 nm of thickness) were measured to confirm the passivation effect on printed SWCNT-TFTs. The mobility (µ_FE_) was calculated from the maximum transconductance (µ_FE_ = 2L(g_m_)^2^/WCi), where Ci and g_m_ are the gate capacitance per unit and transconductance, respectively. The printed dielectric layer’s capacitance of 9 nF/cm^2^ was obtained from the capacitance–voltage measurement method and used as the gate capacitance here. The printed p-type SWCNT-TFTs showed a carrier mobility of 0.010~0.015 cm^2^/Vs, Vth of +10~+12.0 V, and Ion/Ioff ratio of ~10^3^. For the n-type pSWCNT-TFTs, we obtained carrier mobility of 0.003~0.010 cm^2^/Vs, Vth of −7.0~−9.0 V, and Ion/Ioff of ~10^2.5^. Because of the low efficiency of the n-type converting process via the R2R gravure printing, some n-type pSWCNT-TFTs showed low mobilities. To confirm the passivation effect at the ambient conditions, the samples with and without passivation were monitored for 7 days, as shown in Figure 2b,c. The ∆Vth was more noticeable for the n-type pSWCNT-TFT because exposure to air would directly influence the electron carriers. For the trapped charge density calculation in pSWCNT-TFTs, a simple equation (∆Vth = eNtr(t)/Co) was adopted, where e is the elementary charge, Ntr is the trapped charge density, and Co is gate capacitance [24]. 

The ∆Vth of 1.73 V and 7.25 V were, respectively, observed for p-type and n-type pSWCNT-TFT samples without a SiNx passivation layer for 7 days (Figure 2c). On the other hand, SiNx-passivated pSWCNT-TFTs showed stable transfer characteristics for 7 days, as shown in Figure 2b. The passivated sample has the ∆Vth of 0.64 V of and 3.20 V for p-type and n-type under ambient conditions for 7 days. They did not change their ∆Vth value for 3 days, showing a good passivation effect for p-type and n-type pSWCNT-TFTs. The Vth shift was more noticeable for the n-type pSWCNT-TFTs than the p-type ones after the passivation. Thus, the different Vth shifts can be attributed to the adsorption of water and oxygen molecules from the surroundings. That is why the trapped charge density was found to be, respectively, 2.25 × 10^9^/cm^3^ and 6.52 × 10^9^/cm^3^ for passivated and non-passivated p-type pSWCNT-TFT samples, as shown in Figure 2d,e. The trapped charge density of the non-passivated p-type pSWCNT-TFT showed three times higher values than the SiNx-passivated one. Similarly, the n-type pSWCNT-TFT had a trapped charge density of 1.8 *×* 10^8^/cm^3^ and 4.1 × 10^8^/cm^3^ with and without SiNx passivation layers, indicating the 50% suppressing effect. Non-passivated pSWCNT-TFT samples generally had a larger ∆Vth and higher trapped charge density than passivated samples. Furthermore, Figure 2f shows the ∆Vth for three conditions: without the passivation layer, with the passivation layer of CYTOP, and with the passivation layer of SiNx for 7 days. The non-passivated n-type pSWCNT-TFT sample showed a larger ∆Vth value (±0.50 V for p-type and ± 1.7 V for n-type) than the passivated samples. By passivating the single CYTOP layer with 800 nm of thickness, the ∆Vth was reduced to ±1.0 V for p-type and n-type samples from those of the non-passivated one. By comparing the three conditions, the pSWCNT-TFTs with the passivation layer of SiNx showed the most stable characteristics among them. In particular, those results confirmed that the passivation of SiNx on the SWCNT semiconducting layer could improve the pSWCNT-TFT stability by attenuating the permeation of water and oxygen molecules from surroundings so that the trapped charge density was consequently reduced [7]. Moreover, pSWCNT-TFTs passivated with SiNx measured the transfer characteristics under mechanical stress (bending test) for the flexible electronics application. It was found that the transfer characteristics of the p-type under an outer bending radius of 2.5 cm showed a similar property with the non-bending pSWCNT-TFT shown in Appendix A. At the same time, the SiNx-passivated n-type pSWCNT-TFT was also tested for its electrical properties with and without the bending (radius of 2.7 cm) and showed similar electrical properties to the non-bending devices, as shown in Appendix A. This result showed the practical potential of R2R gravure-printed devices to be employed in real fields.

To confirm the stability, the operation test of the pSWCNT-TFT-based inverter with a commercial resistor was carried out under ambient conditions of 70% humidity and 23 °C for one hour. SiNx-passivated p-type pSWCNT-TFTs showed a 0.1 V change in Vp-p (peak-to-peak) output value and shifted slightly upward during continuous operation for 1 h, as shown in Figure 3a. In addition, the inverting property of the n-type pSWCNT-TFT was also measured for 1 h to check the stability under ambient conditions (Figure 3b). The stable inverting property of the n-type pSWCNT-TFT was observed as well. In this case, the Vp-p output decreased by only 0.3 V and shifted slightly downwards. On the other hand, non-passivated samples showed that the signal was shifted up quickly for both p-type and n-type pSWCNT-TFTs even in a short time (approximately 10 min), as shown in Appendix A. To further test the stability of a simple integrated logic gate, five series of CMOS-like inverters, the R2R-printed and passivated SWCNT-TFT-based ring oscillators (1-bit code generator) were fabricated, as shown in Figure 3c. Three samples of 1-bit code generator (non-passivated, CYTOP-passivated, and SiNx-passivated sample) were tested under ambient conditions.

Figure 3d shows the output characteristics of a bare (non-passivated sample) printed 1-bit code generator for 3 h of running time, while Figure 3e shows the output characteristic of a SiNx-passivated 1-bit code generator, tested under ambient conditions. It was found that the non-passivated 1-bit code generator shrank by 28% from the initial Vp-p output by continuously running for 3 h under ambient conditions. On the other hand, the SiNx-passivated 1-bit code generator showed only a 4% change from the initial Vp-p output for 3 h of running time. Based on the stability test on the printed 1-bit code generator, the non-passivated printed 1-bit code generator was more shrunken and shifted in output voltage than the passivated one, because of the trapped charges caused by water and oxygen molecules. Thus, the SiNx layer can effectively block the permeation of water and oxygen molecules to provide device stability. For testing the long-term stability of the SiNx-passivated 1-bit code generator, we also monitored the output characteristics, including frequency and Vp-p of the output signal, and found good stability under ambient conditions for 5 days, as shown in Appendix A.

The standard method (IEC 60,068) for the environmental reliability test for Si-based devices is an 85/85 (85% of humidity at 85 °C) test, a Temperature Humidity Test. Thus, the R2R-printed and passivated SWCNT-TFT-based ring oscillator also tested its stability under the 85/85 test condition. For the 85/85 testing, both passivated p-type and n-type pSWCNT-TFTs with the CYTOP and the SiNx passivation were kept inside the 85/85 chamber to see the environmental reliability. Then, we measured its transfer characteristics every hour. By investigating the ∆Vth and trapped charge density after the 85/85 test, the ∆Vth of the SiNx-passivated p-type pSWCNT-TFT was 0.8 V while it was 3.7 V for the n-type, as shown in Figure 4a. On the other hand, the CYTOP-passivated p-type pSWCNT-TFT showed 5 V of ∆Vth in 3 h while the n-type pSWCNT-TFT changed to the p-type property quickly and completely lost its n-type property, shown as a block dotted line in Figure 4b. Finally, we monitored the output characteristic of a device with and without the SiNx-passivated 1-bit code generator under the 85/85 test. As shown in Figure 4c, the device maintained almost the initial output signal after the 85/85 test for 24 h. However, the non-passivated and CYTOP-passivated 1-bit code generators showed no output characteristics under the 85/85 test (see Appendix A). Those results from the 85/85 test demonstrated that SiNx passivation by R2R PECVD is an efficient way to provide the practical stability of the printed device. 

## 4. Conclusions

In summary, we examined the influence of the SiNx passivation layer on the R2R-printed p-type and n-type SWCNT-TFTs via a combination of R2R gravure printing and R2R PECVD methods. We found that the SiNx-passivated p-type and n-type pSWCNT-TFTs showed only ±0.50 V of threshold voltage change under ambient conditions for 3 days and ±1.20 V of threshold voltage change for 3 h under the Temperature Humidity Test (85% of humidity at 85 °C,85/85). Furthermore, under the 85/85 test conditions, the SiNx-passivated 1-bit code generator survived even after 24 h. By passivating SiNx on the 1-bit code generator, constructed by printing five p-type and five n-type pSWCNT-TFTs, we proved that blocking the permeation of water and oxygen molecules is a way to provide the practical stability of the printed devices. Hence, this work will soon support opening a way to commercialize the R2R-printed SWCNT-TFT-based devices on flexible electronics applications.

## Figures and Tables

**Figure 1 nanomaterials-13-00559-f001:**
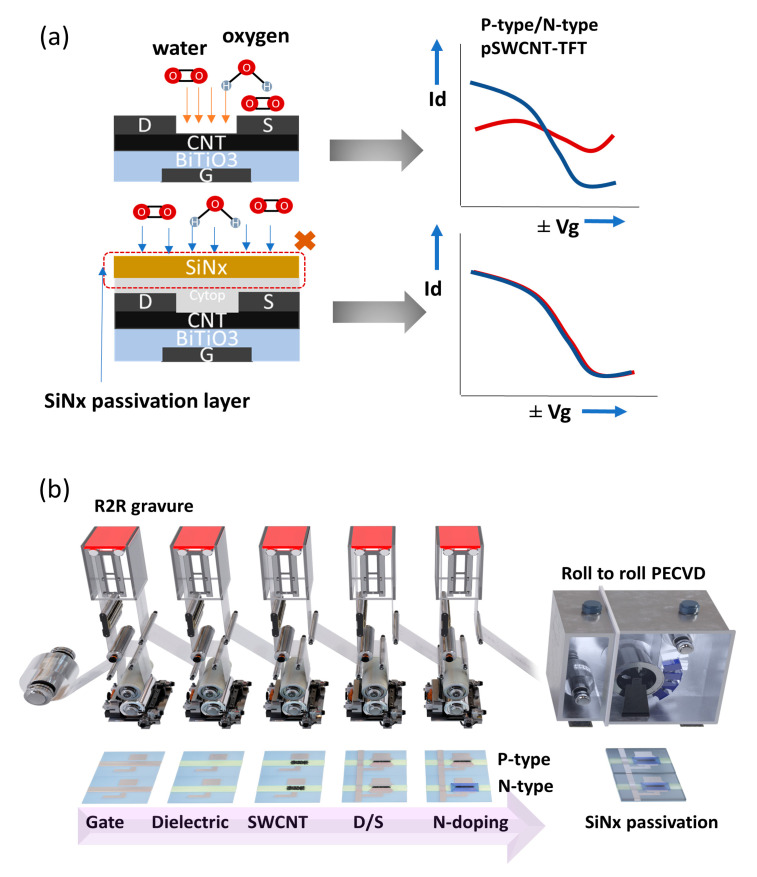
The overall scheme of the role of R2R gravure and R2R PECVD system to continuously deposit a SiNx passivation layer after R2R printing the pSWCNT-TFTs. (**a**) Schematic diagram showing the structure of pSWCNT-TFTs with a passivation layer to prevent the oxygen/water molecules from reaching the pSWCNT-TFT properties illustrated with transfer characteristics before (blue) and after (red). (**b**) Overall R2R gravure printing and R2R PECVD processes for manufacturing the passivated pSWCNT-TFT-based devices.

**Figure 2 nanomaterials-13-00559-f002:**
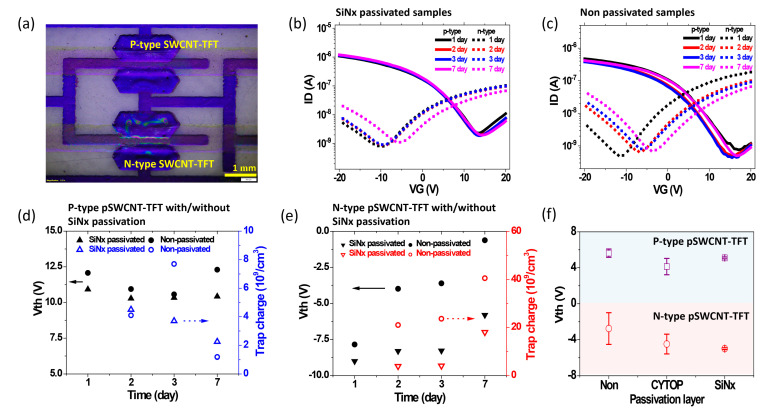
Single pSWCNT-TFT device characterization before/after SiNx passivation process. (**a**) Optical image of SiNx-passivated p-type and n-type pSWCNT-TFTs. Transfer characteristics of R2R-printed p-type and n-type SWCNT-TFTs (**b**) with passivation and (**c**) without SiNx passivation (200 nm thickness). The plot of Vth and trapped charge density variations with and without SiNx passivation layer of (**d**) p-type and (**e**) n-type pSWCNT-TFTs as a function of time. (**f**) Vth variation graph of bare (non-passivated), CYTOP-passivated, and SiNx-passivated p-type and n-type pSWCNT-TFTs.

**Figure 3 nanomaterials-13-00559-f003:**
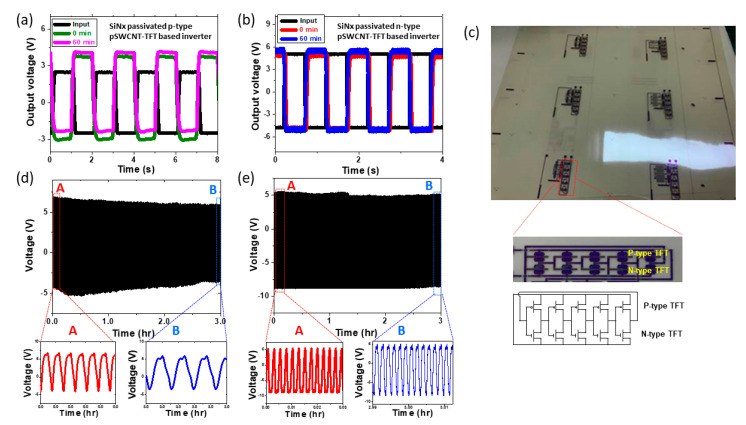
Stability test of pSWCNT-TFTs and 1-bit code generator. Inverting property with SiNx passivation layer of (**a**) p-type pSWCNT-TFT and (**b**) n-type pSWCNT-TFT along with time. (**c**) Optical image of the R2R-printed 1-bit code generator and their circuit design consisting of five p-type and five n-type pSWCNT-TFTs. Stability of 1-bit code generator (**d**) without SiNx passivation layer and (**e**) with SiNx (~200 nm) passivation layer along with time.

**Figure 4 nanomaterials-13-00559-f004:**
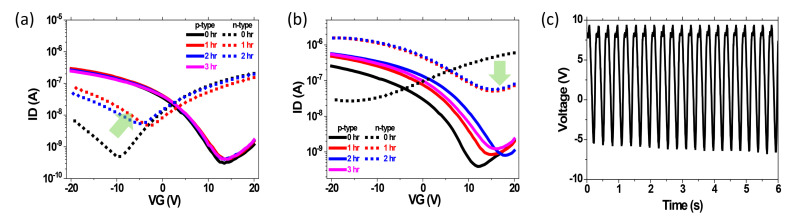
The Temperature Humidity Test (85% humidity at 85 °C) for the printed 1-bit code generator. Transfer characteristics of pSWCNT-TFTs at 85% of identity at 85 °C (**a**) with SiNx passivation layer and (**b**) with CYTOP passivation layer along with time. (**c**) The output characteristics of SiNx (~200 nm)-passivated 1-bit code generator after keeping under 85/85 test condition for 24 h.

## Data Availability

Data can be obtained on request to isinu7@skku.edu. They may also be accessed from the Appendix A after publication of the manuscript.

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
