# Peer review of "Improving Stability of Roll-to-Roll (R2R) Gravure-Printed Carbon Nanotube-Based Thin Film Transistors via R2R Plasma-Enhanced Chemical Vapor-Deposited Silicon Nitride"

_nanomaterials, 2023, doi:10.3390/nano13030559_

Round 1

Reviewer 1 Report

Line 196~197

The thickness of SiNx is 200nm to avoid the cracks. It is better to discuss the experiment results of this paper to show the mechanical stability of SiNx film used as passive layer of device.  And please comment on the flexibility of the device with SiNx passive layer.

Figure 4

Is the figure caption correct for figure 4(a) and (b)? It conflicts with the text of the paper. It seems the Figure 4(a) is for SiNx and (b) for CYTOP.

Author Response

Reviewer #1 (Remarks to the Author):

  1. Reviewer 1 recommended adding more explanation about the mechanical stability of the SiNx passivated layer. Thank you for the valuable comment. As a mechanical stability test, different thickness SiNx films were bent at 10 mm radius with frequency 1 Hz and 10000 cycles, and we checked the cracks on the films. Thickness higher than 200 nm showed cracks on the film's surface after the bending test. These results regarding optical images before and after the bending test were added to the supplementary. Please refer to the revised manuscript (page 5, lines 197-204 and Figures S3 in the revised supplementary).

An outer bending fatigue test has been performed to examine the possibility of cracks on SiNx films as a mechanical stability test. SiNx was coated with different thicknesses on a PET substrate. Then, an outer bending fatigue test was conducted with a radius of 10 mm, under a frequency 1 Hz, and 10,000 cycles. Among different thicknesses of SiNx layer, 300 and 400 nm showed bending failure after 3000 and 1000 cycles respectively, whereas 200 and below 200 nm completed the test without failure, as shown in Figure S3. That's why 200 nm was selected for the passivation.

Figure S3. Optical images of before and after bending test for SiNx layer on PET with thickness (a) 400 nm (bending test for 1,000 cycles) , (b) 300 nm (bending test for 3,000 cycles) and, (c) 200 nm (bending test for 3,000 cycles).

Reviewer 1 has pointed out the Figure 4. Caption conflict between a and b. Thank you for pointing out this issue. The caption has been changed between Figure 4(a) and (b) to validate the text of the paper. Please refer to the revised manuscript on page 9, line 329.

Reviewer 2 Report

The authors present a R2R PECVD method to use SiNx to passivate the pSWCNT FETs, which is pottentially very interesting and important to the TFT applications of CNT. However, before made the decision for publication, the authors must address the following questions and comments:

1. How do you transfer the pSWCNTs from ambient pressure to low-pressure PECVD system with all R2R process?

2. The authors have demonstrated the effect of gas, moisture barrier properties of the SiNx layer. However, for the flexible electronic application as the authors claimed, the readers need to know how stable is pSWCNT FET with the SiNx layer under dynamic bending test.

3. There are typos in the manuscript. Such as :line 130: "detects" should be "defects". The authors should check the manuscript again.

4. line 103, what does "bias stress" mean? electrical bias or mechanical bias? Authors should write it clear.

5. Fig. 1a, the curves don't have clear legend. Which color is which? Are the curves experimental or illustrated?

6. Ref. 23 and Ref. 25 are the same, which is inapproprite self-citation.

Author Response

Reviewer #2 (Remarks to the Author):

  1. Reviewer 2 is concerned about the transfer process of pSWCNTs from ambient pressure to low-pressure PECVD with all R2R processes. Thank you for the valuable comment. R2R gravure printing and R2R PECVD process were carried out separately because, after the R2R gravure process completion, we need to coat the CYTOP to avoid collision issues between energetic electrons and molecules sources of gases. Thus, R2R gravure printing and R2R PECVD experiment were performed separately in this work. However, the R2R gravure and R2R PECVD would be integrated to build up a continuous in-line manufacturing process.

  1. Reviewer 2 recommended discussing the experiment results of the flexibility of the device with SiNx passivation layer. Thank you for indicating to us one of the significant benefits. For the mechanical stability test under the bending status, the devices were bent with a radius 2.5 cm for n-type and 2.7cm for p-type to check the electrical property. The transfer characteristics before and after bending showed similar electrical properties for both p-type and n-type pSWCNT-TFTs. We have added the TFTs characteristic curve in supplementary Figure S4 and discussed the electrical properties before and after the bending test. We have briefly mentioned this in the revised manuscript (page 7, lines 266-274).

Moreover, pSWCNT-TFTs passivated with SiNx measured the transfer characteristics under mechanical stress (bending test) for the flexible electronics application. It was found that the transfer characteristics of p-type under the outer bending radius 2.5 cm showed a similar property with non-bending pSWCNT-TFT shown in Figure S4a. At the same time, the SiNx passivated n-type pSWCNT-TFT was also tested the electrical property with and without the bending (radius of 2.7 cm) and showed similar electrical properties to the non-bending devices, as shown in Figure S4b. This result showed the practical potential of R2R gravure printed devices to be employed in real fields.

Figure S4. Transfer characteristics of SiNx passivated (a) n-type and, (b) p-type pSWCNT-TFTs before and after bending tests.

  1. Reviewer 2 recommended changing the word from ‘detects’ to ‘defects’. Thank you for pointing out an error in the sentence. The word ‘detects’ from line 130 (page 3) was changed to ‘defects’.

  1. Reviewer 2 recommended writing a clear meaning of bias stress, whether it was related to mechanical or electrical one. Thank you for your valuable comment to clear up the meaning. Printed transistors are unstable during the electrical bias at ambient conditions because of the oxygen and water molecules present in the environment. Thus, to show the electrical bias stress stability with SiNx passivation, we performed inverting test experiment under continuous electrical bias stress. Hence, to make it clear to the readers, "electrical" was added in front of the bias stress in the manuscripts. Please refer to the revised manuscript, page 3, line 103.

  1. Reviewer 2 was curious about Figures 1a curves because of not having clear legend as well as color. Thank you for indicating clarity of the overall scheme. The curves show the effectiveness of the SiNx passivation on pSWCNT-TFTs against the oxygen/water molecules. The purpose of the curve was to show the before and after introducing the ambient condition. The appropriate caption is added to mention the purpose of that curve. Please refer to the revised manuscripts, page 4, line 171. Furthermore, regarding for the color issue, we used blue and red colors respectively before and after introducing the TFTs to oxygen/water molecules. Please refer to the revised manuscripts, Fig. 1(a) page 4 (line 171).

  1. Reviewer 2 was confused because the same article was cited in a different number. Thank you for enlightening the mistake in the citation. The citation is rearranged to avoid the conflict, and Ref. 23 and Ref. 25 were removed and put in Ref. 23, page 11 (lines 431-433).

In the revised manuscript, the corrections and additional explanations with new references are accentuated with red to facilitate the review. We hope that the responses above sufficiently address the reviewers’ comments and meet your expectations.

Please contact us at your earliest convenience if you need any further information.

Round 2

Reviewer 2 Report

The current version is good for publishing.